# An Improved Time Delay Measurement Method for the Long-Distance Underwater Environment

**DOI:** 10.3390/s23084027

**Published:** 2023-04-16

**Authors:** Ruisheng Wu, Yuzhe Wang, Lidong Huang, Zhuoyang Zou, Bin Wu

**Affiliations:** 1School of Cyberspace Security, Hainan University, Haikou 570228, China; 2College of Biomedical Engineering & Instrument Science, Zhejiang University, Hangzhou 310027, China; 3Ocean College, Zhejiang University, Hangzhou 316021, China

**Keywords:** long distance, correlation algorithm, underwater time delay, optimal time windows, low SNR, Bellhop model

## Abstract

With the development of underwater navigation and underwater communication, it remains difficult to obtain time delay measurements after propagating long distance. This paper proposes an improved high-accuracy time delay measuring method for long distance underwater channel propagation. First, by sending an encoded signal, the signal acquisition is carried out at the receiving end. Then, to improve signal to noise ratio (SNR), bandpass filtering is carried out on the receiving end. Next, considering the random changes in the underwater sound propagation channel, a strategy is proposed to select the optimal time window for cross-correlation. Then, new regulations are proposed to calculate the cross-correlation results. To verify the effectiveness of the algorithm, we compared it with other algorithms under low SNR conditions using Bellhop simulation data. Finally, the accurate time delay is obtained. With underwater experiments over different distances, the method proposed by the paper achieves high accuracy. The error is about 10^−3^ s. The proposed method makes a contribution to underwater navigation and communication.

## 1. Introduction

Underwater acoustic time delay measurement is widely used in underwater navigation and communication [1,2,3]. The method based on encoded correlation estimation is one of the most important methods with the advantages of high precision and speed. The shallow water environment in the ocean is quite complex and random, which leads to the different characteristics of an underwater acoustic channel, such as long multipath extension, obvious Doppler effect, and limited available bandwidth. With the interference of ocean background noise, it is difficult to detect correlation peaks and correct time delay. In addition, the accuracy of time delay estimation is quite low. Many kinds of time delay estimation algorithms have been created for the shallow water environment and the ocean noise suppression. There are three conventional time delay estimation algorithms for underwater measurement, including the cost function algorithm [4,5,6,7,8], feature structure algorithm [9], and the generalized cross-correlation (GCC) algorithm [10].

The basic principles of time delay estimation are to establish cost function and optimize the optimal delay estimation based on the iterative operation. There are lots of related methods such as the maximum likelihood method [11], the expectation maximization (EM) method [12], and the nonlinear least squares method [13]. However, these methods’ disadvantages are that they need a lot of time to obtain higher estimation accuracy in a low signal-to-noise ratio environment.

Feature structure algorithm is the application of spatial spectrum estimation technology in the field of time delay detection. Many kinds of algorithms are used commonly, including MUSIC [9], ESPRIT [10], and maximum entropy spectral estimation algorithms [11]. This kind of algorithm has high delay resolution and continuous delay estimation, but its excellent delay estimation performance is still limited to signals with flat or nearly flat spectrum.

GCC algorithm was proposed early [12,13], and it was proven to have good robustness and low computational cost, which makes it widely used in underwater acoustic ranging. However, the inference in underwater environments reduces the signal-to-noise ratio, which decreases the performance of the GCC. Improving the signal-to-noise ratio by reducing noise can greatly improve the accuracy of delay estimation. Therefore, plenty of scholars have combined the generalized cross-correlation delay method with other signal processing methods. Stéphenne [14] applied the cepstral filtering technique before GCC. Benesty [15] applied the multichannel cross-correlation coefficient (MCCC) to improve the accuracy of the time delay estimation for reverberation and noise and also demonstrate that accuracy increases with the number of sensors. Vesperini [16] took the neural network provided by the GCC-PHAT (generalized cross-correlation phase transform) mode and adopted a three-stage optimization procedure to find the best network configuration and GCC-PHAT patterns combination. Brandstein [17] employed a signal-dependent criterion to design a GCC filter suitable for both additive noise and multipath conditions. Based on the above research, it can be seen that improving the signal-to-noise ratio can effectively improve the accuracy of delay estimation.

A key technology in GCC algorithm is correlation peak detection. In the long distance and low SNR environments, the correlation peak of the GCC algorithm is often difficult to detect automatically. Peak detection has already developed lots of different methods in the field of signal processing, such as window-threshold techniques [18], wavelet transform [19], Hilbert transform [20], and convolutional neural networks [21,22]. For the peek detection in underwater acoustic ranging, the window length and threshold value should be chosen properly in order to apply to the ocean conditions and the characteristics of the measured signal. The relevant peaks can be made clear by improving the weight function. The effective weights mainly include GCC-PHAT weighting [23], PHAT-ργ weighting [24], and GCC-PHAT-ργ weighting [25].

The false peaks brought by the multipath effect will bring a lot of uncertainty at low SNR. The time delay of the pulse peak related to the signal in the cross-correlation function varies linearly with the distance between the hydrophones. Therefore, Li et al. [26] obtain accurate multipath delay estimates by delaying and aligning the pulse peaks related to the signal, and then coherently summing them. Sun et al. [27] proposed a direct signal classification network based on DT, which selects direct signals from the multipath channel response of a single transmission signal. Liu et al. [28] used a cyclic frequency estimator to estimate the unknown cyclic frequency of the interfering signal and removed the interference from the received signal, leaving only the desired signal, and then estimated using a cross-correlation function.

The contribution of this paper is that we propose a method that can find the correlation peaks accurately. This method is based on the optimal time windows algorithm and new regulations to find the correct peak, which can achieve more accurate time delay estimation in long-distance low SNR environments.

The rest of this paper is organized as follows: First, Section 2 shows the improved correlation for long-distance underwater environment measurement, which includes the new regulations to find the correct peak of correlation and the optimal time windows algorithm. In addition, this section introduces the total flow of this algorithm. Then, Section 3 presents simulations to verify the validity of the proposed method. Section 4 presents the underwater environment measurement experiment of different distance and the discussion. Finally, the conclusion is drawn in Section 5.

## 2. The Improved Time Delay Measurement Method

### 2.1. Hydroacoustic Channel and Signal Model

Based on the encoded correlation principle, this section proposes the improved correction method for underwater time delay measurement of long distance. The encoded signal of measurement is described as follows:(1)$$Y_{1}(t)=S\left(t\right)$$
where *S*(*t*) denotes the encoded signal without background noise.

Sound waves propagate well underwater at an average speed of 1500 m/s. The speed of sound depends on temperature, salinity, and pressure of the seawater. When the signal is propagating close to the ocean surface, the temperature and pressure of the ocean water can be considered constant. Therefore, the propagation velocity in this layer can be considered as a constant value. Sound waves will be absorbed in the process of propagating, resulting in propagation loss. Propagation loss includes spread loss, absorption loss, and scattering loss. The total propagation loss can be represented as:(2)$$A\left(l,f\right)=\alpha \left(f\right)l+\beta {log}_{10}(l)$$
where f denotes the frequency of the signal, l denotes the propagation distance, β denotes the coefficient of spread loss, with value between 1 and 2, and $\alpha \left(f\right)$ denotes the coefficient of absorption loss. After the encoded signal has been transmitted over long distances, considering only the direct path, the received signal is represented as follows:(3)$$Y_{2}(t)=A\left(l,f\right)S\left(t-\tau \right)+N\left(t\right)$$
where τ denotes the time delay of long-distance propagation and *N*(*t*) denotes the background noise. The underwater environment is complex and the multipath of the underwater channel is serious. Sound wave propagating in the ocean is reflected by the seabed, the surface of the sea, and other obstacles. Sound refracts in underwater channels because of the horizontal and vertical sound speed variations. It will also be refracted because of the density in homogeneous media in the ocean. Then, the received signal can be represented as:(4)$$Y_{2}(t)=\sum_{k=1}^{N}a_{k}S\left(t-\tau_{k}\right)+N_{k}\left(t\right)$$
where N denotes the number of paths, $a_{k}$ denotes propagation loss of each paths, $\tau_{k}$ denotes the propagation time of each paths, and $N_{k}\left(t\right)$ denotes independent and equally distributed noise received by each paths. Based on the correlation principle, the correlation between the encoded signal and the received signal can be described as follows:(5)$$\begin{array}{c} R_{12}\left(\sigma \right)=E\left(Y_{1}\left(t_{1}\right)Y_{2}\left(t_{2}\right)\right)=E\left(\sum_{k=1}^{N}a_{k}S\left(t-\tau_{k}\right)S\left(t\right)+\sum_{k=1}^{N}S\left(t\right)N\left(t\right)\right)\\ =\frac{1}{2\pi }\left[\int Y_{1}^{*}\left(w\right)Y_{2}\left(w\right)e^{j\omega \sigma }dw\right] \end{array}$$

Ideally, the direct wave has the shortest distance, which leads to less propagation loss. Therefore, the propagation time can be calculating using the correlation peaks. However, the direct wave may not propagate to the receiver because of the terrain. This problem was confirmed before the experiment in Section 3.

### 2.2. BPSK Signal Model

To obtain a better correlation result, the encoded signal was designed for this section. For long distance communication in water, carrier modulation is widely used. The sinusoidal signal contains three parameters: amplitude, frequency, and phase. The binary phase-shift keying (BPSK) signal model is digitally modulated for data transmission by changing the carrier phase. In BPSK, a binary ‘0′ is represented by one phase of the carrier signal, while a binary ‘1′ is represented by a phase shift of 180 degrees. The BPSK signal model is defined as follow:(6)$$s\left(t\right)=\left\{\begin{array}{c} Asin\left(\omega_{c}t+\theta_{0}\right),0\leq t\leq T\\ Asin\left(\omega_{c}t+\theta_{1}\right),otherwise \end{array}\right.$$
where A denotes the amplitude of signal, $\omega_{c}$ denotes the angular frequency of signal, and $\theta_{0}$ and $\theta_{1}$ denote the phase shifts of 0 bits and 1 bits, respectively. As shown in Figure 1, the BPSK signal model has high resolution, and it is chosen as the encoded signal in this paper.

BPSK signal is resistant to noise and interference, making it suitable for use in low SNR environments. Furthermore, it has relatively low bandwidth requirements compared to other modulation schemes, making it suitable for applications with limited bandwidth. Based on its simplicity, robustness, and low bandwidth requirements, BPSK signal is chosen as the encoded signal in this paper.

### 2.3. Baseline

We choose GCC algorithms as baselines to judge the improvement of the correlation method.

GCC filters the signal before performing a cross correlation analysis of the signals. The correlation between the encoded signal and the received signal can be described as:(7)$$R_{12}=E\left[Y_{1}\left(t\right)Y_{2}\left(t\right)\right]=\int_{0}^{\pi }G_{12}\left(\omega \right)e^{i\omega \tau }d\omega$$
where $G_{12}\left(\omega \right)$ denotes the mutual power spectrum function of $Y_{1}\left(t\right)$ and $Y_{2}\left(t\right)$. Assume that $H_{1}$ and $H_{2}$ is filter function. After the filtering of $Y_{1}\left(t\right)$ and $Y_{2}\left(t\right)$, the mutual spectrum of the output signal can be written as:(8)$$G_{g12}\left(\omega \right)=H_{1}\left(\omega \right)H_{2}^{*}\left(\omega \right)G_{12}\left(\omega \right)$$
where $H_{2}^{*}\left(\omega \right)$ denotes the conjugated matrices of $H_{2}\left(\omega \right)$. Therefore, GCC functions of the encoded signal $Y_{1}\left(t\right)$, and the received signal $Y_{2}\left(t\right)$ can be expressed as:(9)$$R_{g12}^{\left(g\right)}\left(\tau \right)=\int_{0}^{\pi }\psi_{g}\left(\omega \right)G_{12}\left(\omega \right)e^{i\omega \tau }d\omega$$
where $\psi_{g}\left(\omega \right)=H_{1}\left(\omega \right)H_{2}^{*}\left(\omega \right)$, and $\psi_{g}\left(\omega \right)$ represents generalized frequency domain weight components. In the time delay measurement, the correlation peaks will be more obvious by selecting the appropriate weight function. The main weight function includes a Roth processor, smoothed coherence transform (SCOT), and PHAT. GCC-PHAT is used as the baseline in this study, and the function can be expressed as:(10)$$\psi \left(\omega \right)=\frac{1}{\left|G_{12}\left(\omega \right)\right|}$$

### 2.4. The Optimal Regulation for Low SNR

Based on the signal of high signal-to-noise, it is easy to obtain the time delay of long-distance and correlation features. However, in most cases, underwater environments are complex and random. When the SNR is lower, it is hard to obtain the correlation characteristic of encoded signal and received signal. To achieve the aim of long-distance measurement, this section proposes an optimal regulation to obtain the correct result of time delay. This new regulation proposed by this paper is mainly based on the natural characteristics of the underwater environment, which includes continuity of change, energy density distribution, and energy amplitude.

First, the fluid medium changes randomly with time, and its change is slow and continuous, so the continuity of correlation in a short period needs to satisfy the following relation:(11)$$R_{12}^{\prime }\left(\sigma_{n}\right)>0\;or\;R_{12}^{\prime }\left(\sigma_{n}\right)<0$$
where $R_{12}^{\prime }\left(\sigma_{n}\right)$ denotes the derivative of the correlation energy, which is used to calculate the continuity of correlation. As the positivity and negativity of $R_{12}^{\prime }\left(\sigma_{n}\right)$ has no contribution to the value of correct correlation, the first factor $f_{1}$ in the optimal regulation can be define as follows:(12)$$f_{1}=\zeta_{1}\max\left|{R^{\prime }}_{12}\left(\sigma_{n}\right)\right|$$
where $\zeta_{1}$ denotes the weight factor. To measure the standard value of correct correlation, the dimensions and units of the data have no contribution to the right answer and may cause the offset of accurate value. Therefore, the factors in the evaluation criterion need to be normalized to range [0, 1]. Equation (12) can be rewritten as follows:(13)$$f_{1}\left(\sigma_{n}\right)=\frac{\zeta_{1}\left|{R^{\prime }}_{12}\left(\sigma_{n}\right)\right|}{\max\left|{R^{\prime }}_{12}\left(\sigma_{n}\right)\right|}$$

Then, based on the energy correlation characteristics, these peaks with strong correlation energy are distributed intensively. In addition, correlation peaks should meet the principle of concentrated distribution of probabilities, as follows:(14)$$N_{R_{12}}\to max||-3\delta \leq N_{R_{12}}\leq 3\delta$$
where $N_{R_{12}}$ denotes the number of peaks distributed near the correlation peak, and *δ* denotes the average of correlation peak in a short period. The second factor $f_{2}$ in the optimal regulation can be define as follows:(15)$$f_{2}\left(\sigma_{n}\right)=\frac{\zeta_{2}N_{R_{12}}}{\max N_{R_{12}}}$$
where $\zeta_{2}$ denotes the weight factor. Finally, the most important point is that compared to other peaks, the correlation energy peak should be kept at a maximum, as follows:(16)$$\left|R_{12}\left(\sigma_{n}\right)\right|\geq \max\left|R_{12}\left(\sigma_{n}\right)\right|$$

The third factor $f_{3}$ in the optimal regulation can be defined as follows:(17)$$f_{3}\left(\sigma_{n}\right)=\frac{\zeta_{3}\left|R_{12}\left(\sigma_{n}\right)\right|}{\max\left|R_{12}\left(\sigma_{n}\right)\right|}$$
where $\zeta_{3}$ denotes the weight factor. Based on three important standards of discrimination for low SNR, we proposed three factors to decide the correct peak of correlation. In this way, the regulation can be written as follows:(18)$$\begin{array}{c} L\left(\sigma_{n}\right)=f_{1}\left(\sigma_{n}\right)+f_{2}\left(\sigma_{n}\right)+f_{3}\left(\sigma_{n}\right)=\frac{\zeta_{1}\left|{R^{\prime }}_{12}\left(\sigma_{n}\right)\right|}{\max\left|{R^{\prime }}_{12}\left(\sigma_{n}\right)\right|}+\frac{\zeta_{2}N_{R_{12}}}{\max N_{R_{12}}}\\ +\frac{\zeta_{3}\left|R_{12}\left(\sigma_{n}\right)\right|}{\max\left|R_{12}\left(\sigma_{n}\right)\right|} \end{array}$$
where $L\left(\sigma_{n}\right)$ denotes the standard value of correct correlation, and ${0\leq \zeta }_{1}$, $\zeta_{2}$, $\zeta_{3}\leq 1$ and $\zeta_{1}+\zeta_{2}+\zeta_{3}=1$. The function $L\left(\sigma_{n}\right)$ takes the maximum at value of correct correlation, and the side lobe should be lower. $\zeta_{1}$, $\zeta_{2}$, and $\zeta_{3}$ have different values because of different encoded signals and different marine environments. It is necessary to fit the best value using the simulation data. Assume that the number of simulation data at the same distance in the same environment is M. The optimal value of the three parameters can represent a nonlinear least squares problem whose objective function can be expressed as follows:(19)$$\begin{array}{c} \underset{\zeta_{1},\zeta_{2},\zeta_{3}}{\max}F\left(\zeta_{1},\zeta_{2},\zeta_{3}\right)=\sum_{i=1}^{M}[L(\sigma_{c})-\frac{1}{N}\sum_{j=1}^{N}L(\sigma_{n})]\\ =\sum_{i=1}^{M}[f_{1}\left(\sigma_{c}\right)+f_{2}\left(\sigma_{c}\right)+f_{3}\left(\sigma_{c}\right)\\ -\frac{1}{N}\sum_{j=1}^{N}\left(f_{1}\left(\sigma_{n}\right)+f_{2}\left(\sigma_{n}\right)+f_{3}\left(\sigma_{n}\right)\right)] \end{array}$$
where $\sigma_{c}$ denotes time delay of direct wave, and N denotes the number of sampling points for the data. To obtain the optimal parameters, the following steps are iterated:Step 1: Assign the initial value to
$\zeta_{1,0}$,
$\zeta_{2,0}$, and
$\zeta_{3,0}$.Step 2: For $k^{th}$ iteration, calculate the Jacobian matrix:
(20)$$F^{\prime }\left(\zeta_{1,k},\zeta_{2,k},\zeta_{3,k}\right)=\left[\begin{array}{c} \frac{\partial F}{\partial \zeta_{1}}\\ \frac{\partial F}{\partial \zeta_{2}}\\ \frac{\partial F}{\partial \zeta_{3}} \end{array}\right]$$The parameters are updated using the Jacobian matrix:(21)$$\left[\begin{array}{c} \zeta_{1,k+1}\\ \zeta_{2,k+1}\\ \zeta_{3,k+1} \end{array}\right]=\left[\begin{array}{c} \zeta_{1,k}\\ \zeta_{2,k}\\ \zeta_{3,k} \end{array}\right]-F^{\prime }\left(\zeta_{1,k},\zeta_{2,k},\zeta_{3,k}\right)F(\zeta_{1,k},\zeta_{2,k},\zeta_{3,k})$$Step 3: The increments
$∆\zeta_{1,k}$, $∆\zeta_{2,k}$, and $∆\zeta_{3,k}$ can be calculated, and, when they are all small enough to $10^{-5}$, stop iterating.Step 4: Otherwise, return to Step 2.

### 2.5. Adaptive Optimal Time Window

This section presents an efficient algorithm to select an optimal time window for the correlation algorithm. The longer the time window length, the larger the gain, making the correlation peak sharper. However, the ocean environment is not constant.

In the process of adaptively selecting the length of the time window, the ratio of correlation peak height to the side lobe height should be measured to determine the optimal time window.

The optimal time window algorithm steps based on the maximum SNR are as follows:Step 1: Starting with the starting point of received signal, take the data with a window length of
L=l, and do cross-correlation with the encoded signal. Calculate the ratio of the correlation peak to the side lode as $R\left(L\right)$.Step 2: Change the window length, take L=2l, L=3l,…, until L≤N (*N* is the length of the received signal) and repeat step 1.Step 3: Find the optimal window length at the starting point.Step 4: Repeat step 1–step 3 for the different starting points of the received signal and find the best starting point and the optimal window length.

### 2.6. The Process of the Algorithm

This method proposed by this paper is used to solve the problem of the time delay measurement of low SNR. Due to complexity and randomness of the underwater environment, the SNR of the underwater signal becomes very chaotic. When the SNR deteriorates, it is very difficult to measure the time delay of long distance. So, based on the new method for low SNR, long-distance measurement can be achieved. The specific flow chart of the algorithm is shown as Figure 2.

The paper calculates the correlation result to make the long-distance measurement of underwater by combining the encoded signal with the received signal. First, the received signal need to be band-pass filtered, which can improve the SNR. Then, based on the change of SNR and the underwater environment, it is important to select the best time window to conduct the correlation algorithm. Due to the underwater background noise and multipath effects of the underwater environment, there are lots of interference peaks. To find the correlation peak, the paper proposes a new strategy to choose the peaks. Based on the process above, we can obtain high precision time delay after long-distance propagation underwater.

## 3. Simulation

For this section, data with different SNR are simulated using Bellhop model to verify the effectiveness of the proposed method and compare the performance with CC and GCC-PHAT.

### 3.1. Data Generation

The simulation in this section is performed using the Bellhop model. Bellhop is an underwater acoustic simulation software used for calculating propagation loss in ocean acoustics. It is based on ray tracing and wave theory methods. Bellhop can simulate sound propagation in various ocean environments, including different seabed terrains, temperature, and salinity distributions. These are the experimental conditions set for using the Bellhop model in this section:The speed of sound is constant throughout the ocean.The gaussian white noise is used as the ocean background noise.The signal used for simulation is BPSK signal model, which was introduced in the previous section.The seabed terrains are set up the same as the experimental sea area in Section 4.

Assume the moment of signal emission is zero. The main parameters of the simulation are shown in Table 1.

First, Bellhop is used to simulate the propagations from source to receivers at 50 km, and received signal can be calculated by convolution. Then, Gaussian white noise with a bandwidth of 8000 is added to the received signal to obtain data with different SNR at 50 km. The SNR is calculate using the following formula:(22)$$SNR=10log\frac{\sum_{0}^{T}{\left|A\left(l,f\right)S\left(t-\tau \right)\right|}^{2}}{\sum_{0}^{T}{\left|Y_{2}(t)\right|}^{2}}$$

Finally, the performance of different methods with different SNR will be compared in next subsection.

### 3.2. Simulation Results

This subsection analyzes the performance of different approaches using data with different SNR. The correct time delay is obtained by correlating the encoded signal and the data with the original signal without adding noise on the receiver. In high SNR, all three methods can get the delay accurately. However, with the decrease in the SNR, the time delay calculated by every method begin to deviate. Figure 3 shows the correlation results of the three different methods in −40 dB at the distance of the 50 km, and we calculated the errors of different methods in low SNR as shown in Table 2.

When the SNR is low, GCC-PHAT can not get an approximate time delay. Traditional cross correlation and our proposed method can get the precise delays in −35 dB. When the SNR is lower, our proposed method obtains the results more accurately than the traditional cross correlation.

## 4. Experiment and Discussion

The section presents the experiment to verify the effectiveness of the method proposed by this paper. It is mainly divided into two parts; one part describes the experiment for evaluating underwater channel over different distances, and the other one presents a detailed discussion analyzing the results. 

### 4.1. Verification Experiment

This subsection introduces different long-distance experiments of underwater measurement, which include 10 km, 20 km, 30 km, and 50 km. The verification experiment uses the BPSK signal as encoded signal, and the emission and reception are both at 100 m underwater. The mean value of sound speed at a depth of 100 m during the experimental period is about 1500 m/s, and it is used for the delay calculation below. The distance between the emission and reception points are obtained by GPS. Due to the long distance and complexity of the underwater environment, the SNR of measurement is lower at a longer distance. However, the method proposed in this paper can still get a correct time delay. As shown in Figure 4, a sound source was used to send an encoded signal, and a receiver was used to receive the acoustic signal at a different distance. Due to the complexity of the underwater environment, the sound source is deployed in the middle of the sea, while the hydrophones are deployed on the seabed. The distance between the sound source and the hydrophone sensors is changed by adjusting the position of the ship equipped with the source.

In the post-processing step, the optimal time windows are determined by the calculating the SNR in different periods. As shown in Figure 5, the correlation peaks between the encoded signal and the received signal are obtained based on the optimal time windows. At the same time, the multipath effect and Doppler effect always exists in underwater environments. Therefore, there might be two or more correlation peaks after propagation. In this case, we use the method proposed by this paper to find the correct main peak of time delay. In addition, influenced by the underwater background noise, there are many false peaks around the main peak. Therefore, the new regulations proposed by this paper are used to find the correct peaks. Finally, we can obtain the correct time delay via the method proposed by the paper in a low SNR underwater environment.

Figure 6 shows the time delay of 10 km in underwater environment. Due to the relatively short distance, the SNR is high. Influenced by the underwater ambient noise, there are lots of interference peaks around the true time delay peak. We can find the correct time delay is 6.668 s, which is close to the theoretical value of 6.666 s, which is obtained in Section 3.2.

Figure 7 shows the time delay of 20 km in underwater environment. We can obtain the correct time delay is 13.297 s via the method proposed by this paper, which is also close to the theoretical value of 13.333 s. 

The time delay of 30 km is shown in Figure 8. Due to the farther distance, the attenuation of the encoded signal gets severer, and the SNR is lower. With the underwater random noise, there are lots of interference peaks around the real time delay peak. We can obtain the correct time delay is 20.013 s via the proposed method, which is also close to the theoretical value of 20.000 s.

Finally, the time delay of 50 km is shown in Figure 9. It is the farthest distance tested in this paper, the attenuation of encoded signal gets the worst, and the SNR is the lowest. In the case of the lowest SNR, we can get the correct time delay is 33.329 s via the method proposed by this paper, which is still close to the theoretical value of 33.333 s. 

Based on these results, we can obtain the correct time delay of underwater propagation close to the theoretical value, which proves the effectiveness of the proposed methods.

### 4.2. Comparison of Experiment

To show the superiority of the method proposed by this paper, we conducted four sets of comparison experiments. As shown in Table 3, we can find that the method proposed by this paper have higher accuracy in the underwater environment. In comparison, the traditional correlation and GCC-PHAT have weak ability in underwater environment. In addition, for measurements of different distances, their measurement is far away from the theoretical calculated value. Based on comparison, the method proposed by this paper is effective in time delay measurement of underwater long distance.

### 4.3. Discussion

In this paper, we have conducted different distance experiments to verify the effectiveness of this method. First, we can obtain the high accuracy of measurement results for 10 km, 20 km, 30 km, and 50 km via four sets of experiments, and their time delay error are 0.002 s, 0.036 s, 0.013 s, and 0.004 s, respectively. Second, the attenuation is different in different distances, and the propagation loss increases with increasing distance. So when the distance is 30 km and 50 km, the SNR is lower, which makes it relatively difficult to find the correct peak. In this case, the regulations proposed by this paper can solve this problem and find the real time delay peak under ocean ambient noise, which verifies the effectiveness of this method. In addition, when conducting the 50 km experiment, the SNR is the lowest, and it is much hard to find the time delay peak. However, the method proposed by this paper can accurately measure the time delay. Finally, the method proposed by this paper can do better than the traditional correlation and GCC-PHAT according to the comparison experiment. In this section, the proposed method is proven to be an effective method of time delay estimation by the data collected in the experiment.

## 5. Conclusions

This paper proposes a new method based on the correlation algorithm to solve the measurement problem of long-distance underwater transmission in low SNR conditions. The proposed method optimizes the adaptive time window to ensure the optimal SNR to calculate the time delay result in underwater environments. Then, the proposed method is verified by simulation data using the Bellhop model, and it shows good robustness even under low SNR. Finally, based on the different distance measurement experiments and comparison experiments, the proposed method has better performance than the traditional correlation and GCC-PHAT, which verified the effectiveness and superiority of this approach is effective in a practical oceanic environment. The proposed method can overcome the complexity and randomness in long-distance transmission in the ocean environment, which contributes to underwater navigation and underwater communication.

## Figures and Tables

**Figure 1 sensors-23-04027-f001:**
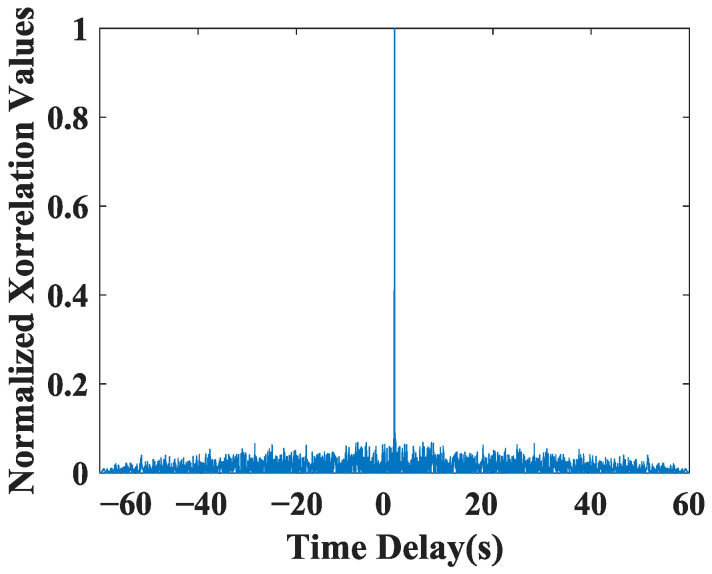
Correlation results of the BPSK signal model.

**Figure 2 sensors-23-04027-f002:**
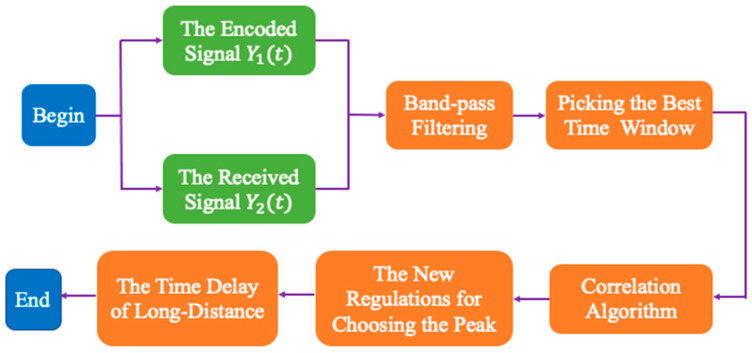
The flow chart of the improved algorithm.

**Figure 3 sensors-23-04027-f003:**
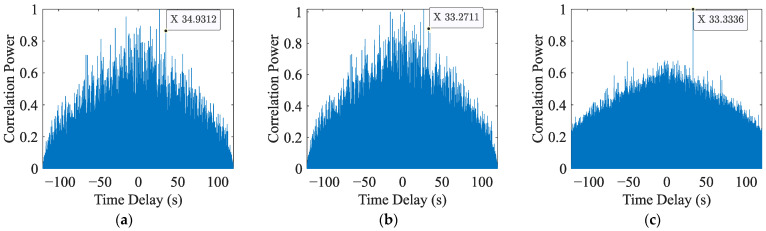
Correlation results of the three different methods in −40 dB. (**a**) Correlation results of GCC-PHAT after normalization in −40 dB, (**b**) correlation results of CC after normalization in −40 dB, and (**c**) correlation results of our method in −40 dB.

**Figure 4 sensors-23-04027-f004:**
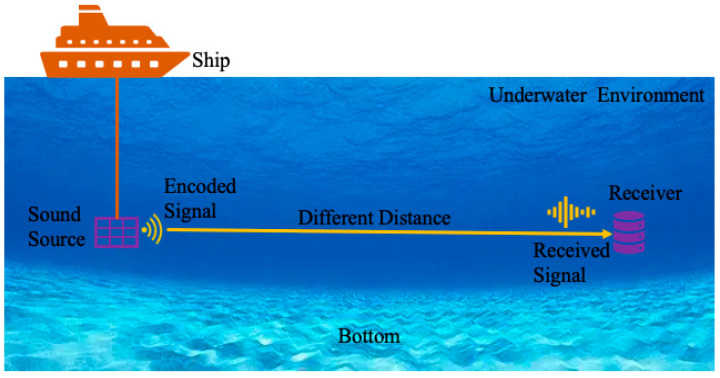
The flow chart of the experiment.

**Figure 5 sensors-23-04027-f005:**
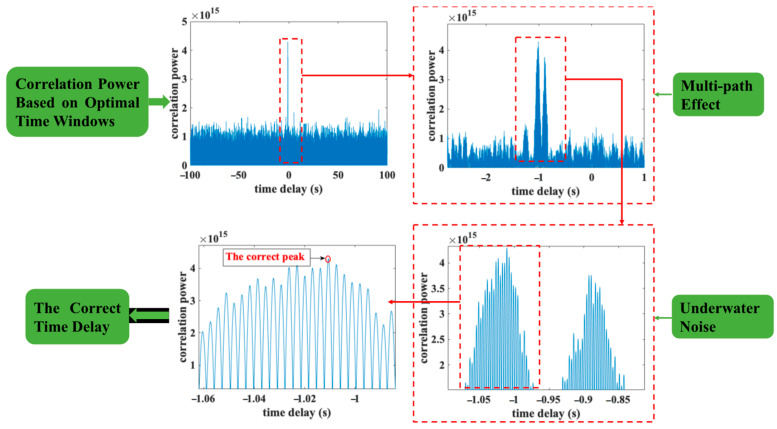
The flow chart of post processing.

**Figure 6 sensors-23-04027-f006:**
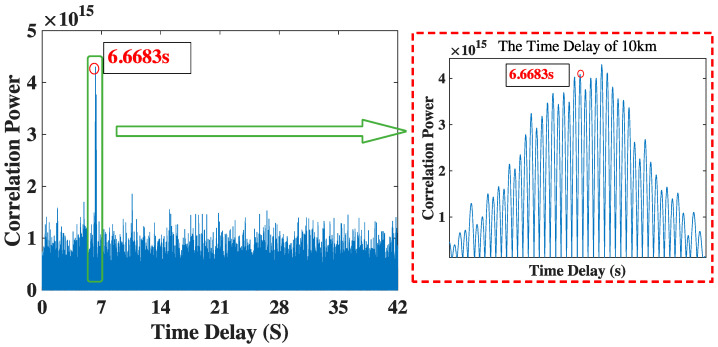
The time delay result of 10 km.

**Figure 7 sensors-23-04027-f007:**
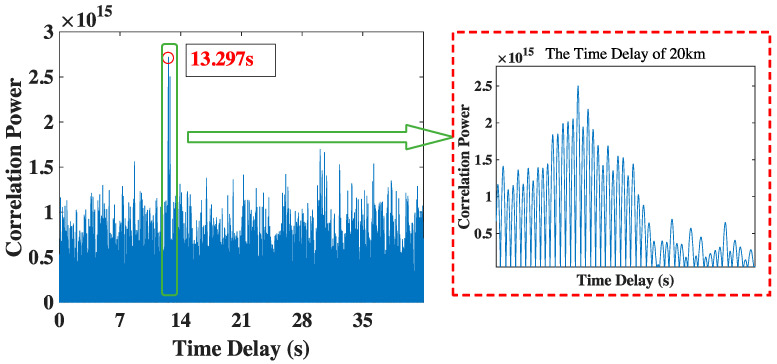
The time delay result of 20 km.

**Figure 8 sensors-23-04027-f008:**
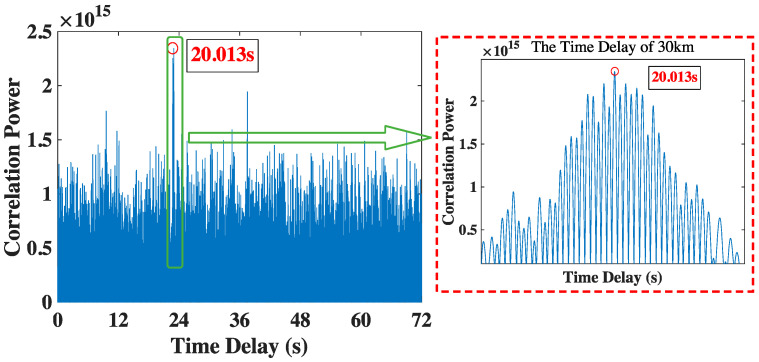
The time delay result of 30 km.

**Figure 9 sensors-23-04027-f009:**
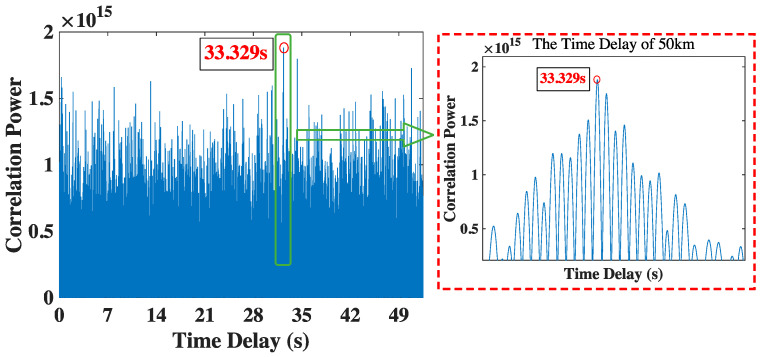
The time delay results of 50 km.

**Table 1 sensors-23-04027-t001:** Input parameters of the simulation.

Parameter	Value	Unit
Sound of speed	1500	m/s
Frequency of sampling	16,000	Hz
Signal frequency	200	Hz
The duration of sampling	120	s
Number of sources	1	-
Depth of sources	100	m
Depth of receivers	100	m
Number of beams	10,001	-
Distance	50	km

**Table 2 sensors-23-04027-t002:** Time delay errors in low SNR.

SNR	GCC-PHAT	CC	Our Method
−30 dB	0.0003	0.0361	0
−35 dB	1.4154	0.0003	0
−40 dB	1.5979	0.0622	0.0312

**Table 3 sensors-23-04027-t003:** The comparison experiment between the method and other method.

Experiment Distance	Theoretical Value	GCC-PHAT	CC	The Method Proposed by this Paper
10 km	6.666 s	7.467 s	6.727 s	6.6683 s
20 km	13.333 s	13.978 s	13.431 s	13.297 s
30 km	20.000 s	20.957 s	20.052 s	20.013 s
50 km	33.333 s	32.068 s	33.499 s	33.329 s

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
