# Peer review of "An Improved Time Delay Measurement Method for the Long-Distance Underwater Environment"

_sensors, 2023, doi:10.3390/s23084027_

Round 1

Reviewer 1 Report

Expand the list of keywords. Give names and recommendations for equipment.

The article contains more comments and errors that need to be eliminated. The authors should work on the comments.

The list of references is insufficient for such topics. The authors should review more deeply and seriously. Sources should also be from high-level magazines. Show which of the researchers worked on this problem. The authors present a lot of mathematics, but do not give enough justification. Sections 4.3 and 5 are very poor. Section 5 does not hold water at all. It is necessary to substantiate the conclusions and prospects, give values ​​and comparative analysis. Conclusions should correspond to the tasks and goals set by the authors.

Author Response

Thank you very much for your extremely important opinion! Here is my response for your comments.

Comment 1: Expand the list of keywords. 

Response : We are so greatful for your kind suggestion. We have already expended the list of the keywords and you can find in Line 22.

Comment 2: Give names and recommendations for equipment.

Response : We are so greatful for your kind suggestion. Our verification experiment is based on secrecy, so names and recommendations for equipment  may can not be provided. But we add some experimental information in Line 301 to Line 305 to make the experimental setup more complete.

Comment 3: The list of references is insufficient for such topics. The authors should review more deeply and seriously. Sources should also be from high-level magazines. Show which of the researchers worked on this problem. 

Response : We are so greatful for your kind suggestion. We do have a lot of shortcomings in the introduction and we have corrected them.

Comment 4: The authors present a lot of mathematics, but do not give enough justification.

Response : We are so greatful for your kind suggestion. This method is not a formula derived from theoretical deduction, but rather a set of factors that we discovered through long-term experimentation and experience, which can help obtain more accurate delay results. We have repeatedly checked the formulas and supplemented the parameter iteration in Line 212.

Comment 5: Sections 4.3 and 5 are very poor. Section 5 does not hold water at all. It is necessary to substantiate the conclusions and prospects, give values ​​and comparative analysis. Conclusions should correspond to the tasks and goals set by the authors.

Response : We are so greatful for your kind suggestion. We conducted new simulation experiments using Bellhop because we believed that the previous simulation experiments were not rigorous enough. In the conclusion section, we summarized the results of the new simulation experiments and the experimental results, demonstrating the effectiveness of the proposed method in both simulation and real-world experiments. We also proved the contribution of the proposed method to underwater navigation and communication.

Reviewer 2 Report

The manuscript presents a method to improve accuracy of measuring time of flight from a source to a receiver in underwater channel using BPSK signal. Results with simulation and experimental data were shown with the proposed method and other two commonly used methods, CC and GCC-PHAT to compare performance at different ranges. It seems that the proposed method can predict the travel time better than the other two under the experimental conditions with an assumed sound speed at 1500m/s with an accuracy to 10-3 s. It however did not provide any supported information on the experiment for this sound speed, or any other information that can be used to derive the sound speed at the experimental site.

The manuscript needs to be improved to correct many errors as in the list below (it is not necessarily comprehensive and complete). More importantly, the detail descriptions of the simulation and the experiment set up should be included. The information is necessary to make assessment of the results.

List of corrections:

Line 13: replace make the signal energy more concentrated with improve signal to noise ratio

Line 61: insert the before above research

Line 90: use an instead of a

Line 94: delete ß

Line 95&98: replace expansion with spread

Line 100: replace way with path

Line 104: sound refracts in underwater channel because of the sound speed variations in horizontal and vertical

Figure 1: more details the signals used for the correlation

Line 160: change (5) to (12)

Eq (6): is it correct?

Line 181: change directive to direct, and if to of

Eq (19) change ?? in  ?3|?12(??)|max|?12(??)| to ?c

Line 220: replace concentrate the power of encoded signal with improve the signal to noise ratio.

3. Simulation

Is the simulation in an infinite water medium?

Line 233:  replace evenly distributed with constant

Table 1: signal frequency?

Line 242: analyses

Line 242: replace the first using with of

Line 242: approaches

Line 245: replace in with with SNR=

Line 245: replace in with at

Figure 3 (a): It would be helpful to explain the reason of the large decrease of correlation around t=60 with GCC- PHAT

Line 257: replace conducts with presents

Line 264: add to the last of the line with at longer distances

4.1 Verification experiment: more information about the measurement should be given, for example: the time and the date of the measurements, water depth, sound speed profile, bathymetry along the transmission path, seabed properties, signal used, how the distances were determined (GPS? Its accuracy?), how the timing of transmitting and receiving signal was measured, etc.

Author Response

We sincerely thank you for your careful reading of our paper and for providing so many valuable suggestions for revision. Thank you very much for pointing out the grammar errors we carelessly made. We have corrected them according to your suggestions. Therefore, I will only reply to your questions and comments here.

Comment 1 : It however did not provide any supported information on the experiment for this sound speed, or any other information that can be used to derive the sound speed at the experimental site. More importantly, the detail descriptions of the simulation and the experiment set up should be included. The information is necessary to make assessment of the results.

4.1 Verification experiment: more information about the measurement should be given, for example: the time and the date of the measurements, water depth, sound speed profile, bathymetry along the transmission path, seabed properties, signal used, how the distances were determined (GPS? Its accuracy?), how the timing of transmitting and receiving signal was measured, etc.

Response : We are so greatful for your kind suggestion. We think your comments are both about the insufficient of our experimental conditions, so we put your comments together.

Sorry for the oversight  and we have added the missing content for experiment set at Line 301 to Line 305.

The signal used is the BPSK signal and we determined the distances based on the GPS. The mean value of sound speed at a depth of 100m during the experimental period is about 1500m/s. Our verification experiment is based on secrecy, so other detail information about the measurement may can not be provided but I can ensure the authenticity of the experimental data.

We also complete the parameters of the simulation part in Table 1.

Comment 2 : Is the simulation in an infinite water medium?

Response : We are so greatful for your question. Your question have made us rethink our simulation. The previous simulation used the simple ray tracing, which was not rigorous enough for multi-path. The simulation was redone using the Bellhop model, resulting in more convincing data. The results obtained are different from those submitted in the previous manuscript. After a resimulation, my answer to your question is in a water layer of a finite path.

Reviewer 3 Report

This work presents an interesting approach for underwater acoustic time delay measurement. It is a progression from the previous work in the field with some novelty in it. However, there are several aspects that need to be addressed by the authors as the current form lacks some important information, which means a considerable amount of work is still needed to improve the quality of the manuscript so that it can be sufficient for acceptance.

Major:

1.    Most of the references discussed in the introduction section seem to be quite outdated, while the time delay estimation and measurement for underwater application has been extensively studied. Thus, the authors should provide discussion of more recent works in the field to prove that the work proposed and discussed in this manuscript are the state of the art with novelty in it.

2.    Equation 19 indicates the objective function is to get the maximum of the right-hand side terms, but in line 186, the process is to get the minimum from the object function, is this correct?

3.    In line 187, what threshold is considered as small enough?

4.    In line 184, for the optimal set of parameters, how those increment parameters are adjusted during the iteration? Step 2 should provide more detail on this.

5.    In line 191, after checking section 3, it seems there is no calculation process shown for those weighting parameters as mentioned here.

6.    For section 2.5, the optimal time window will be set based on the signal input. However, this optimal value will only be applicable to the specific dataset, and it will probably become invalid if the data characteristics changes during the receiving and processing stages, which may be a normal scenario as the ocean environment is unpredictable. Also, the algorithm presented here looks very inefficient as it basically is a trial-and-error approach to loop through all possible values, which is computationally expensive, so I’m wondering how it can be classified as an “adaptive” algorithm?

7.    In section 3.1, why the assumption of absorption decay is neglected? It is a coefficient related to the propagation loss term directly based on equation 2, the authors should provide convincing explanation of why one is considered while another is not.

8.    How is the time delay error in table 2 calculated?

Minor:

·      Reference 24 and 26 are the same

·      Reference 25 cannot be found online.

·      The authors should put the full name of the terminologies at the places they are firstly introduced, for example, “Binary Phase-shift keying” for BPSK signal model in section 2.2

·      Line 160, do the authors mean equation 12? Also, equation 12 should be revised to have the max term to match with equation 13, otherwise “f_1” cannot be used in both equations.

·      Equation 19 should be expressed in a more compact way as the normalized terms have been introduced in previous equations, thus it is better to use those notations to express this long equation instead of the current form.

·      Writing should be improved; typos are found in several places as well.

Author Response

Comment 1 : Most of the references discussed in the introduction section seem to be quite outdated, while the time delay estimation and measurement for underwater application has been extensively studied. Thus, the authors should provide discussion of more recent works in the field to prove that the work proposed and discussed in this manuscript are the state of the art with novelty in it.

Response : We are so greatful for your kind suggestion. The list of references is insufficient for our topics, so we complemented the introduction.

Comment 2 : Equation 19 indicates the objective function is to get the maximum of the right-hand side terms, but in line 186, the process is to get the minimum from the object function, is this correct?

Response : We are so greatful for your kind question. Sorry that we made a low-level mistake, we have already revised it and write the iteration process more clearly.

Comment 3 : In line 187, what threshold is considered as small enough?

Response : We are so greatful for your kind question. Sorry that we missed it before, we added the threshold at Line 215.

Comment 4: In line 184, for the optimal set of parameters, how those increment parameters are adjusted during the iteration? Step 2 should provide more detail on this.

Response : We are so greatful for your kind suggestion. We complement the iteration from Line 211 to Line 216.

Comment 5: In line 191, after checking section 3, it seems there is no calculation process shown for those weighting parameters as mentioned here.

Response : We are so greatful for your kind suggestion. We complement the calculation process of the parameters in the Section 2 from Line 211 to Line 216.

Comment 6 : For section 2.5, the optimal time window will be set based on the signal input. However, this optimal value will only be applicable to the specific dataset, and it will probably become invalid if the data characteristics changes during the receiving and processing stages, which may be a normal scenario as the ocean environment is unpredictable. Also, the algorithm presented here looks very inefficient as it basically is a trial-and-error approach to loop through all possible values, which is computationally expensive, so I’m wondering how it can be classified as an “adaptive” algorithm?

Response : We are so greatful for your kind question. Because in ocean environments, the channel is time-varying, and it is difficult to simulate the actual channel through simulation or other methods. The method of selecting the window length can maximize the time gain. The adaptive time window in this paper is used to better handle delay calculations for long distances to verify the correlation algorithm we proposed. 

Comment 7 : In section 3.1, why the assumption of absorption decay is neglected? It is a coefficient related to the propagation loss term directly based on equation 2, the authors should provide convincing explanation of why one is considered while another is not.

Response : We are so greatful for your kind question. The previous simulation used the simple ray tracing, which was not rigorous enough for multi-path.
the simulation was redone using the Bellhop model and vertical wave absorption attenuation was added in the course of the Bellhop simulation, resulting in more convincing data. The results obtained are different from those submitted in the previous manuscript.

Comment 8 : How is the time delay error in table 2 calculated?

Response : We are so greatful for your kind question. The correct time delay is obtained by correlating the encoded signal and the data with the original signal without adding noise on the receiver. We added this part in Line 280.

Comment 9 : Reference 24 and 26 are the same.

Response : We are so greatful for your kind suggestion. We deleted Reference 26.

Comment 10 : Reference 25 cannot be found online.

Response : We are so greatful for your kind suggestion. Not sure if it is a Word version problem, Reference 25 can be found at https://www.mdpi.com/1424-8220/22/19/7254.

Comment 11 : The authors should put the full name of the terminologies at the places they are firstly introduced, for example, “Binary Phase-shift keying” for BPSK signal model in section 2.2.

Response : We are so greatful for your kind suggestion. We revised this problem in Line 137 and made some additions to the introduction of the BPSK signal.

Comment 12 : Line 160, do the authors mean equation 12? Also, equation 12 should be revised to have the max term to match with equation 13, otherwise “f_1” cannot be used in both equations.

Response : We are so greatful for your kind suggestion. We revised equation 12.

Comment 13 :  Equation 19 should be expressed in a more compact way as the normalized terms have been introduced in previous equations, thus it is better to use those notations to express this long equation instead of the current form.

Response : We are so greatful for your kind suggestion. We revised equation 19.

Comment 14 :  Writing should be improved; typos are found in several places as well.

Response : We are so greatful for your kind suggestion. We tried our best to improve the writing and revised the typos.

Round 2

Reviewer 1 Report

good corrections made

Author Response

Thank you very much for taking the precious time to read our article again.